# Testing the General Deductive Reasoning Capacity of Large Language Models Using OOD Examples

**Abulhair Saparov**[†]    **Richard Yuanzhe Pang**[†]    **Vishakh Padmakumar**[†]    **Nitish Joshi**[†]

**Seyed Mehran Kazemi**[△]        **Najoung Kim**[△,β,∗]        **He He**[†,∗]

[†]**New York University,** [△]**Google,** [β]**Boston University**
as17582@nyu.edu

## Abstract

Given the intractably large size of the space of proofs, any model that is capable of general deductive reasoning must generalize to proofs of greater complexity. Recent studies have shown that large language models (LLMs) possess some abstract deductive reasoning ability given chain-of-thought prompts. However, they have primarily been tested on proofs using modus ponens or of a specific size, and from the same distribution as the in-context examples. To measure the general deductive reasoning ability of LLMs, we test on a broad set of deduction rules and measure their ability to generalize to more complex proofs from simpler demonstrations from multiple angles: depth-, width-, and compositional generalization. To facilitate systematic exploration, we construct a new synthetic and programmable reasoning dataset that enables control over deduction rules and proof complexity. Our experiments on four LLMs of various sizes and training objectives show that they are able to generalize to compositional proofs. However, they have difficulty generalizing to longer proofs, and they require explicit demonstrations to produce hypothetical subproofs, specifically in *proof by cases* and *proof by contradiction*.

## 1   Introduction

In many tasks that require deductive reasoning, such as theorem proving or medical diagnosis, the complexity of proofs can grow without bound via the use of multiple deduction rules and the composition of subproofs. Given the large space of proofs, it is infeasible to find data to cover proofs of all sizes. Therefore, a general reasoning model must extrapolate to complex proofs from simpler ones. Recent work has shown that LLMs, combined with in-context learning (ICL) and chain-of-thought (CoT) prompting, are capable of deductive reasoning to an extent [Huang and Chang, 2022, Han et al., 2022, Wei et al., 2022b, Kojima et al., 2022, Lewkowycz et al., 2022, Nye et al., 2021, Gontier et al., 2020]. However, much of the prior work focused on a limited set of deduction rules such as modus ponens [Zhang et al., 2022a, Saparov and He, 2023, Tafjord et al., 2021]. In addition, the evaluation is *in-demonstration*, where the test example comes from the same distribution as the in-context demonstrations. In this work, we evaluate whether LLMs are capable of general deductive reasoning by measuring how well they generalize to proofs that are more complex

---

[∗]Equal contribution.

[†,△]GPT and PaLM experiments in this paper were conducted independently. Authors affiliated with NYU[†] were responsible for the GPT experiments, and authors affiliated with Google[△] were responsible for the PaLM experiments.

37th Conference on Neural Information Processing Systems (NeurIPS 2023).

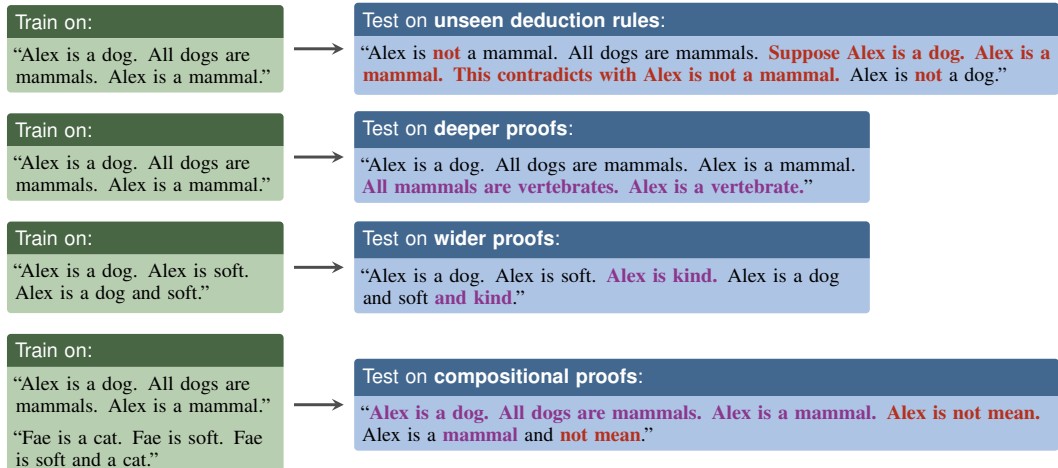

FIGURE 1: An overview of the kinds of OOD generalization that we test in our experiments. Each training example is a sample CoT demonstration provided to the LLM in the few-shot prompt, whereas each test example is a sample proof that the model is expected to output.

than their demonstrations.[1]

We characterize the complexity of proofs from three angles: the deduction rules involved, the depth of the proof (i.e. length of a sequential chain of proof steps), and the width of the proof (i.e. the number of premises of each proof step). Each of the three dimensions contributes to the overall size of the proof. To measure the general deductive reasoning ability of LLMs, we extend prior studies in two key ways. First, we determine whether LLMs have learned a *complete* set of deduction rules, beyond modus ponens. Second, we evaluate whether they can reason over longer proofs than those given as in-context examples (depth- and width- generalization); and whether they are able to use multiple different deduction rules in a single proof (compositional generalization). Figure 1 shows an overview of our study.

Our findings suggest that in-context learning is best applied to reasoning tasks by including examples that cover a diverse set of deduction rules, and keeping the examples simple. The in-context examples should especially contain examples of deduction rules that are less familiar to the model (i.e. proof by cases and proof by contradiction), and distractors should be provided for such examples as the model is more prone to overfitting.

We test four different LLMs of different scales and training objectives: GPT-3.5 175B [Ouyang et al., 2022], PaLM 540B [Chowdhery et al., 2022], LLaMA 65B [Touvron et al., 2023], and FLAN-T5 11B [Chung et al., 2022], and we find:

1. CoT is able to elicit out-of-demonstration (OOD) reasoning in LLMs generalizing to compositional proofs. This is somewhat surprising given the amount of previous work that claim that LLMs are *not* able to generalize compositionally [Hosseini et al., 2022, An et al., 2023]. See Section 4.2.2 and Figure 6.
2. ICL generalizes differently compared to supervised learning (i.e. gradient descent on in-context examples). We find numerous examples where it is strictly worse to provide in-context examples from the same distribution as the test example. For instance, in some cases, we observe better generalization to compositional proofs when the in-context examples each contain individual deduction rules. See Sections 4.2.2 and 4.3, and Figures 6 and 8.
3. However, the LLMs cannot generalize to some deduction rules without explicit demonstrations, specifically, *proof by cases* and *proof by contradiction*, suggesting that pretraining is not sufficient to teach the model to generate hypothetical subproofs. See Section 4.2.1 and Figure 4.
4. Model size does not strongly correlate with performance. Smaller (but not the smallest) models with instruction tuning and longer pretraining perform comparably to larger models.

---

[1]All analysis code, data, data generation scripts, and model outputs are publicly available at github.com/asaparov/prontoqa

| Dataset | Automated evaluation of proofs | Contains proofs with multiple deduction rules | Tests proof depth generalization | Tests proof width generalization | Tests compositional generalization | Data generation code available |
|---|---|---|---|---|---|---|
| CLUTRR Sinha et al. [2019] | ✗ | ∼ | ✗ | ✓ | ∼ | ✓ |
| LogiQA Liu et al. [2020] | ✗ | ✓ | ∼ | ∼ | ∼ | human-annotated |
| ProofWriter Tafjord et al. [2021] | ✓ | ∼ | ✓ | ∼ | ∼ | ✗ |
| FOLIO Han et al. [2022] | ✗ | ✓ | ∼ | ∼ | ∼ | human-annotated |
| PRONTOQA Saparov and He [2023] | ✓ | ✗ | ✓ | ✗ | ✗ | ✓ |
| PRONTOQA-OOD (this dataset) | ✓ | ✓ | ✓ | ✓ | ✓ | ✓ |

TABLE 1: Comparison of existing datasets to evaluate reasoning ability. Datasets marked with ∼ contain examples of varying width, depth, and compositionality, but these are not programmable (i.e. we cannot generate new examples controlling for these variables), and splitting the existing examples would produce highly imbalanced splits.

## 2 Related work

**OOD generalization of LLMs.** Previous work has measured the generalization ability of LLMs on tasks such as bit parity and Boolean variable assignment [Anil et al., 2022], semantic parsing [Hosseini et al., 2022, Qiu et al., 2022], deductive reasoning [Zhang et al., 2022a, Sanyal et al., 2022, Kazemi et al., 2023], and arithmetic reasoning [Kudo et al., 2023], where the length/complexity of the test example is greater than that of the in-context examples. On the bit parity and variable assignment tasks, LLMs are able to generalize to longer inputs with scratchpad prompting [Nye et al., 2021], but this generalization is imperfect, and accuracy still degrades with increasing input length. Generally, larger models tend to be better at generalization than smaller ones. The studies on reasoning were limited to reasoning using modus ponens. Wu et al. [2021] tests the OOD generalization of transformers and graph neural networks on symbolic mathematical reasoning. Our study more systematically examines OOD generalization of LLMs to larger proofs as well as to other deduction rules.

**Evaluating reasoning abilities of LLMs.** A number of recent studies measured the reasoning ability of LLMs [Huang and Chang, 2022, Han et al., 2022]. Table 1 provides a comparison of our proposed dataset to datasets from these studies. Many of these datasets are not amenable to automated evaluation of proofs, relying instead on measuring label accuracy. The datasets also do not test for proof width generalization and compositional generalization. Some datasets focus on a limited set of deduction rules, namely modus ponens. Our work is closest to PRONTOQA [Saparov and He, 2023] but extends it to a complete set of deduction rules and to compositional proofs.

**Understanding in-context learning.** Recent work has shed some light on ICL, and the mechanism by which the model learns from in-context examples. Akyürek et al. [2023], Dai et al. [2023], von Oswald et al. [2022] showed that transformers can learn in-context by performing gradient descent on in-context examples internally. Xie et al. [2022], Wang et al. [2023] show that LLMs exhibit behavior similar to that of topic models where their output is dependent on a latent topic, and the in-context examples help to specify the topic. Ye et al. [2022] demonstrated that ICL is more effective when the in-context examples are both diverse and relevant to the test example. An et al. [2023] and Levy et al. [2022] explored the effect of in-context demonstrations on compositional generalization, showing that it benefits from diverse and individually simple demonstrations. Our results contribute to this growing literature by showing that the generalization behavior in ICL is different from that of supervised learning, and so algorithms such as gradient descent are not the only mechanisms underlying ICL.

## 3 Approach

**A programmable dataset.** Our main evaluation approach is to prompt the LLM with simpler proofs and test it on proofs with greater depth and width, or with those using additional deduction rules. Therefore, we require a programmable approach to data generation, where the deduction rules

| Deduction rule | Formal definition | Natural language example |
|---|---|---|
| Implication elimination (i.e. modus ponens) | $$\frac{f(a) \quad \forall x(f(x) \to g(x))}{g(a)}$$ | "Alex is a cat. All cats are carnivores. Alex is a carnivore." |
| Conjunction introduction | $$\frac{A \quad B}{A \wedge B}$$ | "Alex is a cat. Alex is orange. Alex is a cat and orange." |
| Conjunction elimination | $$\frac{A \wedge B}{A}$$ | "Alex is a cat and orange. Alex is orange." |
| Disjunction introduction | $$\frac{A}{A \vee B}$$ | "Alex is a cat. Alex is a cat or orange." |
| Disjunction elimination (i.e. proof by cases) | $$\frac{A \vee B \quad A \vdash C \quad B \vdash C}{C}$$ | "Alex is a cat or a dog. Suppose Alex is a cat . . . then Alex is warm-blooded. Suppose Alex is a dog . . . then Alex is warm-blooded. Alex is warm-blooded." |
| Proof by contradiction | $$\frac{A \vdash B \quad \neg B}{\neg A}$$ | "Alex is cold-blooded. If Alex is a mammal, Alex is not cold-blooded. Suppose Alex is a mammal. Alex is not cold-blooded. This contradicts with Alex is cold-blooded. Alex is not a mammal." |

TABLE 2: An overview of the deduction rules in PRONTOQA-OOD. The notation $A \vdash B$ denotes entailment: that $B$ is provable from $A$.

```
''Alex is a dog.  All dogs are mammals.  Alex is a mammal.  Alex is
blue.  All mean things are not blue.  Suppose Alex is mean.  Alex is
not blue.  This contradicts with Alex is blue.  Therefore, Alex is
not mean.  Alex is a mammal and not mean.''
```

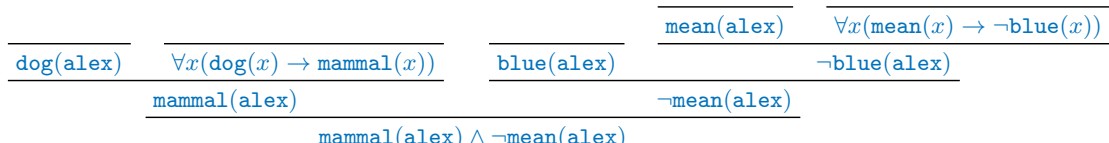

FIGURE 2: An example of a compositional proof containing modus ponens, proof by contradiction, and conjunction introduction, shown in both natural language and a formal tree representation.

used, as well as the depth and width of each proof, are controllable parameters. To this end, we propose PRONTOQA-OOD, a generative process for synthetic reasoning questions. Each example in PRONTOQA-OOD contains a handful of premises, a query (the target fact to be proved/disproved), and a gold CoT containing the proof of the query. See Figure 10 for an example from this dataset. Specifically, we extend the PRONTOQA dataset [Saparov and He, 2023] that contains proofs generated from synthetic world models using modus ponens. (1) To evaluate reasoning using different deduction rules, we generate proofs with deduction rules for all connectives in propositional logic: conjunction ∧, disjunction ∨, implication →, and negation ¬. (2) To study width/depth generalization, the proof depth and width are controllable parameters in proof generation, where they control the number of generated deduction rules, and the number of premises in each deduction rule, respectively. (3) To study compositional generalization, we generate compositional proofs using a simple recursive procedure, where each proof contains multiple subproofs with distinct deduction rules.

**Generating proofs with a complete set of deduction rules.** We follow the deduction rules of *natural deduction* [Gentzen, 1935, Pfenning, 2004], a well-studied proof calculus with desirable completeness properties.[2] Examples of each deduction rule are shown in Table 2. For each type of deduction rule, we randomly generate a proof that applies that rule (see details in section A.4. An example of a compositional proof is shown in Figure 2. To prevent LLMs from exploiting knowledge from pretraining to solve the problems without reasoning, we use fictional names for all concepts (e.g. "wumpus" instead of "cat" etc.).

---

[2]With minor differences: Negation introduction/elimination, truth introduction, and falsity elimination are impossible to express in natural text, and they are omitted. We use proof by contradiction to reason over negation.

**Varying proof width and depth.** To characterize the size or complexity of each proof, we represent each proof as a tree (Figure 2), where each proof step corresponds to a node, and its premises correspond to the parent nodes. Then the size of the proof can be naturally described by the width and depth of this tree. When generating proofs, we control the depth by continuing to append proof steps until a proof of the desired depth is generated. The number of premises of deduction rules is set to the desired proof width.

**Generating compositional proofs.** To generate proofs combining many different types of deduction rules, we use a simple recursive procedure: (1) select a deduction rule uniformly at random, (2) select the premises for the selected rule, (3) recursively generate a subproof for each premise. See section A.5 for details and pseudocode.

**Adding distractors.** One key challenge to OOD generalization is shortcut solutions. For example, given the facts "Alex is a cat," "All cats are feline," since there is no other fact of the form "All cats are...," the model can deterministically follow the only valid deduction and conclude "Alex is feline." To make the heuristics uninformative, we add distractor sentences. In the above case, a distractor sentence would be "All cats are graceful." Then the model is forced to choose the correct premise from two options for the next deduction step. See Section A.7 for details on distractors for all deduction rules.

**Formal evaluation of chain-of-thought.** Unlike previous datasets that evaluate on a binary true/false answer, PRONTOQA-OOD requires LLMs to generate full proofs.[3] Therefore, we need a way to evaluate the correctness of the output proofs directly. The sentences in PRONTOQA-OOD are syntactically simple and amenable to semantic parsing, which allows us to formally analyze each step in the CoT. To determine whether a predicted CoT is correct, we: (1) semantically parse each CoT sentence into first-order logic, (2) determine whether each logical form follows from previous logical forms via a rule of deduction, and (3) compute whether there exists a path of correct steps from the premises to the goal. An example of this process is shown below:

$$
\begin{array}{ccccc}
\text{"Alex is a dog.} & & \texttt{dog(alex)}, & & \\
\text{All dogs are mammals.} & \longrightarrow & \forall x(\texttt{dog}(x) \to \texttt{mammal}(x)), & \longrightarrow & \dfrac{\forall x(\texttt{dog}(x) \to \texttt{mammal}(x)) \quad \texttt{dog(alex)}}{\texttt{mammal(alex)}} \\
\text{Alex is a mammal."} & & \texttt{mammal(alex)} & &
\end{array}
$$

Each proof step is considered correct if it is valid and if it immediately follows one of its premises.[4] For further details see section A.6.

## 4 Results

In this section, we test existing LLMs on PRONTOQA-OOD and analyze whether they can produce longer and compositional proofs given simpler demonstrations. We experiment with a variety of models, with different sizes and training objectives, as shown in Figure 3. In all experiments, we use 8-shot chain-of-thought prompting.[5]

| | FLAN-T5 | LLaMA | GPT-3.5 | PaLM |
|---|---|---|---|---|
| **Model Size** | 11B | 65B | 175B* | 540B |
| **Instruction Tuned** | ✔ | ✖ | ✔ | ✖ |
| **RLHF** | ✖ | ✖ | ✔ | ✖ |
| **Access** | Open | Limited | Limited | Limited |

FIGURE 3: An overview and properties of the LLMs in our experiments. We place an asterisk* for GPT-3.5 since we were not able to verify its size.

We compare performance in two settings: (1) an *in-demonstration* (ID) setting where the 8 in-context demonstrations come from the same distribution as the test example, and (2) an *out-of-demonstration* (OOD) setting where the in-context demonstrations come from a distribution that is different from that of the test example. 95% confidence intervals are provided for all results.

---

[3]True/false queries introduce an extra implicit negation step of the final conclusion. The negated conclusion may be tricky to represent in natural language given the extensive space of potential conclusions of the proofs that we generate.

[4]Note that if we only check for validity, the following proof would be considered correct: "Fae is a cat. All cats are carnivores. Fae is a carnivore. All carnivores are mammals. Fae is a mammal. All mammals are not herbivorous. Fae is not herbivorous." Here, the goal is to prove "Fae is not herbivorous." Every step of the proof is correct except "All mammals are not herbivorous." To avoid this, we require each step to immediately follow one of its premises.

[5]Following PRONTOQA, since adding further in-context examples did not improve performance.

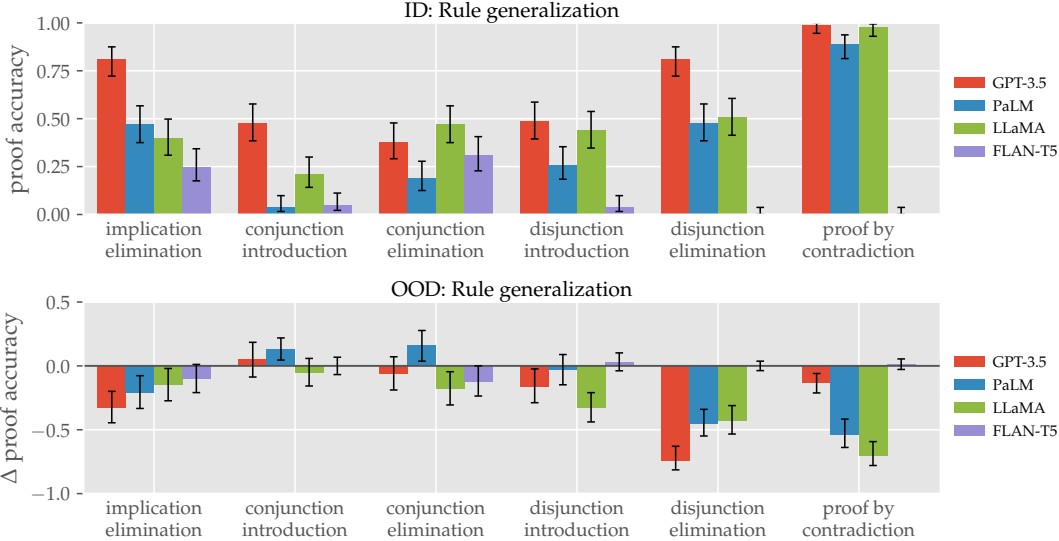

FIGURE 4: **(top)** Proof accuracy across examples with different deduction rules. The in-context examples and test examples come from the same distribution. **(bottom)** Change in proof accuracy, where the test example is out-of-demonstration with respect to the in-context examples. That is, the test example has the specified deduction rule, but the in-context examples are uniformly distributed over *all other* deduction rules. See Figure 11 in the Appendix for the equivalent plot with absolute proof accuracy on the y-axis. See Figure 5 for an incorrect example. Implication elimination examples have proof width of 1 and depth of 2. Conjunction introduction, conjunction elimination, and disjunction introduction examples have proof width 3 and depth 2. Disjunction elimination examples have proof width 3 and depth 1. Proof by contradiction examples have proof width 2 and depth 1.

> *Prove:*  Max is a gorpus.
>
> *Predicted answer:*  Max is a tumpus or a rompus or a lempus.  Max is a tumpus.
> Tumpuses are wumpuses.  Max is a wumpus.  Rompuses are gorpuses.  Max is a gorpus.
> Max is a gorpus.
>
> *Expected answer:*  Assume Max is a tumpus.  Tumpuses are gorpuses.  Max is a gorpus.
> Assume Max is a rompus.  Rompuses are gorpuses.  Max is a gorpus.
> Assume Max is a lempus.  Lempuses are gorpuses.  Max is a gorpus.
> Since Max is a tumpus or a rompus or a lempus, Max is a gorpus.

FIGURE 5: Example of an incorrect proof generated by GPT-3.5 on an out-of-demonstration disjunction elimination example. The premises (axioms) are given in blue, and invalid steps are given in red. For the full example, see Figure 14 in the Appendix.

## 4.1 Can LLMs use deduction rules other than modus ponens?

We first evaluate whether LLMs "know" all deduction rules (Table 2) when provided with corresponding CoT prompts. For each deduction rule, we independently and identically generate 8 in-context examples and one test example, and prompt the model to answer the test example. We run each experiment for 100 trials and measure the accuracy of the output proofs. The accuracies are shown in the top chart of Figure 4. We emphasize that the Δ proof accuracies in the bottom row of the figure should be interpreted in comparison with the accuracies in the top row (e.g. for some rules, the zero Δ accuracy for FLAN-T5 is due to zero absolute accuracy). For clarity, we provide the same plots using absolute accuracy rather than Δ accuracy in Section A.2.

In general, most models are able to use each deduction rule reasonably well, with GPT-3.5 performing the best. Similar to prior studies [Liang et al., 2022], we do not observe a strong correlation between model size and performance. LLaMA performs comparably to PaLM, despite being smaller. FLAN-T5 is smaller and performs reasonably on implication and conjunction elimination, but is not able to learn the other deduction rules.

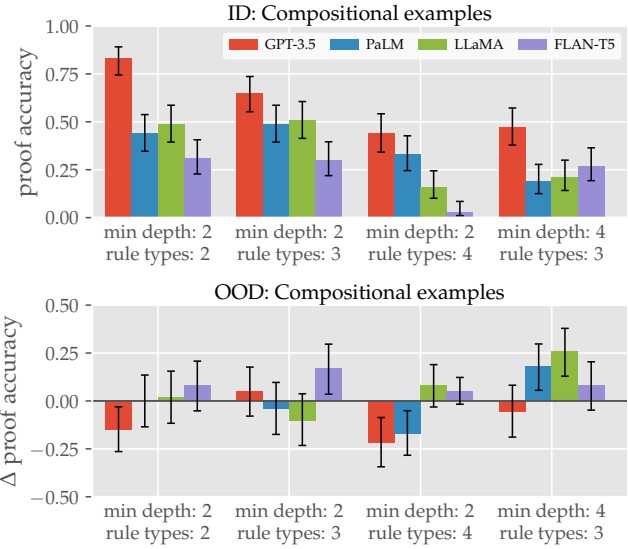

*Prove:* `Polly is not a lempus.`

*Predicted answer:* `Polly is a wumpus, a jompus, and a tumpus. Everything that is a wumpus, a jompus, and a tumpus is not a lempus. Polly is not a lempus.`

*Expected answer:* `Polly is a tumpus. Polly is a jompus. Polly is a wumpus. Polly is a wumpus and a jompus and a tumpus. Everything that is a wumpus, a jompus, and a tumpus is not a lorpus. Polly is not a lorpus. Assume Polly is a lempus. Each lempus is an impus and a lorpus and a rompus. Polly is an impus and a lorpus and a rompus. Polly is a lorpus. This contradicts with Polly is not a lorpus. Polly is not a lempus.`

FIGURE 6: **(top-left)** Proof accuracy on compositional examples where the in-context examples are also compositional examples with the same min depth and number of rule types. **(bottom-left)** Change in proof accuracy where the test examples are compositional but the in-context examples are those of individual deduction rules. See Figure 12 in the Appendix for the equivalent plot with absolute proof accuracy on the y-axis. **(right)** Example of an incorrect proof generated by GPT-3.5 on an out-of-demonstration example with min depth 2 and 4 rule types. The premises (axioms) are given in `blue`, and invalid steps are given in `red`. For the full example, see Figure 15 in the Appendix.

## 4.2 Out-of-demonstration generalization

### 4.2.1 Can LLMs generalize to unseen deduction rules?

While the above results show that LLMs are able to reason with a variety of deduction rules, it is unclear whether the ability is learned from in-context examples or elicited from pretraining. We test the LLM with examples where the test proof requires a deduction rule that does not appear in the in-context examples (i.e. for each in-context example, we sample a deduction rule uniformly at random from the set of deduction rules excluding that of the test example). Our intuition was that LLMs would not be able to use deduction rules unless given explicit demonstrations thereof (aside from those like modus ponens which are well-represented in pretraining). The change in proof accuracy relative to the ID setting is shown in the bottom chart of Figure 4. Evidently, the models are able to use four deduction rules despite not being shown an in-context example with those rules: both conjunction rules, disjunction introduction, and (somewhat) implication elimination. GPT-3.5 was additionally able to use proof by contradiction by relying on an alternate deduction rule called modus tollens (i.e. given $\neg f(c)$ and $\forall x(g(c) \rightarrow f(c))$, conclude $\neg g(c)$). This is in contrast with McKenzie et al. [2022] which showed that reasoning with modus tollens exhibited inverse scaling behavior, and yet GPT-3.5 is able to use it correctly without any demonstrations. However, Wei et al. [2022a] has shown that when trained with additional compute, models are able to perform modus tollens. GPT-3.5 performed worse on disjunction elimination possibly due to the fact that there is no equivalent alternate rule (an example of an error is given in figure 5). However, no model is able to use disjunction elimination and proof by contradiction without demonstrations.

### 4.2.2 Can LLMs generalize to compositional proofs?

Next, we test whether the model is able to generalize to compositional proofs that contain multiple different deduction rules. In the ID setting, the in-context examples and test examples are both generated from the same distribution of compositional proofs. In the OOD setting, the in-context demonstrations contain non-compositional examples of each rule that appears in the test example. In Figure 6, in all but three experiments, we observe that the gap in proof accuracy between the ID and OOD settings is close to zero, indicating that the models are able to generalize compositionally

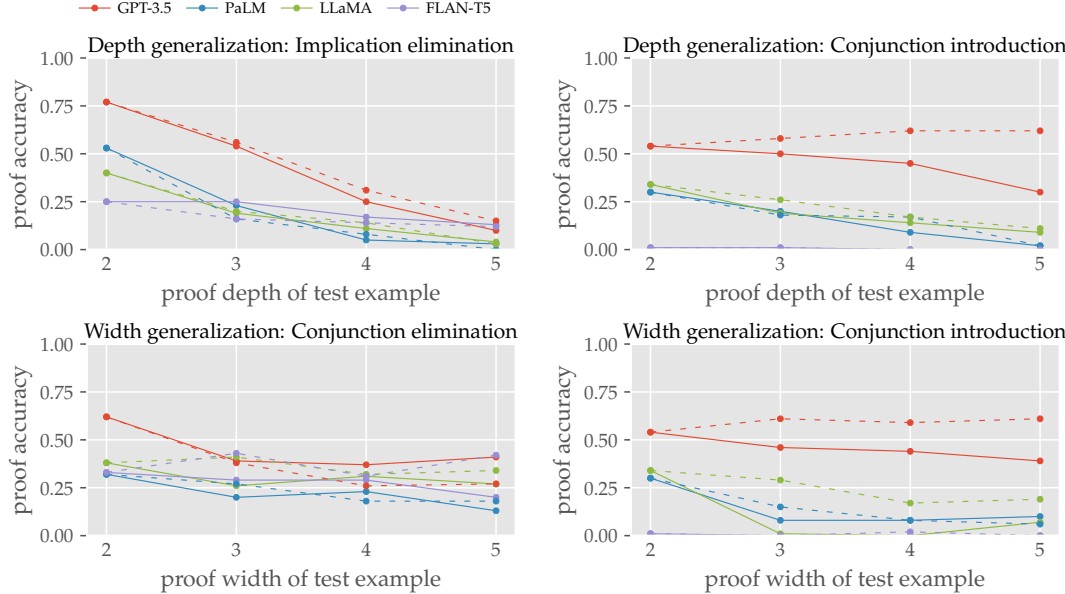

FIGURE 7: **(top row)** Proof accuracy vs proof depth of test examples, where in-context examples have fixed proof depth 2. **(bottom row)** Proof accuracy vs proof width of test examples, where in-context examples have fixed proof width 2. Dashed lines indicate in-distribution accuracy, where the depth and width of the in-context examples are the same as that of the text-examples. In these experiments, there are 4 rather than 8 in-context examples.

to an extent. This is surprising since past studies show that LLMs struggle with compositional generalization, but this could be due to the fact that much of the previous work focused on semantic parsing rather than on reasoning. But there is prior work showing that models can generalize compositionally in some settings, such as in Hosseini et al. [2022] (see Figure 4) and in Press et al. [2022] (see Figure 6). In addition, our study is limited by the token limit of the LLMs, as we are not able to further increase the complexity of the proofs without reducing the number of in-context examples, which would render the results difficult to compare. GPT-3.5 and PALM have difficulty when the number of rule types is 4, with an example of an incorrect output given in the right side of Figure 6. Interestingly, PALM, LLAMA, and FLAN-T5 sometimes perform better in the OOD setting than in the ID setting, showing that, in ICL, it is not always best to provide demonstrations from the same distribution as the test example.

### 4.2.3 Can LLMs generalize to bigger proofs?

To test whether LLMs can generalize to bigger proofs, we test the models on examples where the proof width or depth is larger than those of the in-context examples. As is evident from Figure 7, when shown demonstrations of proofs of depth 2, the models' performance decreases with increasing depth. But this is due to the increase in the inherent difficulty of the task, as both ID and OOD accuracies decrease with increasing depth. Though the notable exception is GPT-3.5 on conjunction elimination, where ID accuracy remains high as OOD accuracy decreases. Models are able to generalize better with increasing proof width on conjunction elimination, possibly because there are ample examples of long conjunctions in natural language, but only GPT-3.5 is able to generalize to greater proof widths on conjunction introduction.

### 4.3 Do distractors help OOD generalization?

In supervised learning, one challenge to OOD generalization is spurious correlations [Zhang et al., 2022b]. Intuitively, if ICL were to behave like supervised learning on in-context examples [Akyürek et al., 2023, Dai et al., 2023, von Oswald et al., 2022], we would expect that without distractors, the models would overfit to the in-context examples and perform poorly on OOD examples. An example where distractors hurt generalization is shown in Figure 9 where GPT-3.5 copies many of the distractor sentences into the output, likely due to the fact that it has learned to apply a copying heuristic from the in-context demonstrations. It seems only GPT-3.5 acquires these heuristics in implication and disjunction elimination. Surprisingly, this is not the case for all deduction rules, as is

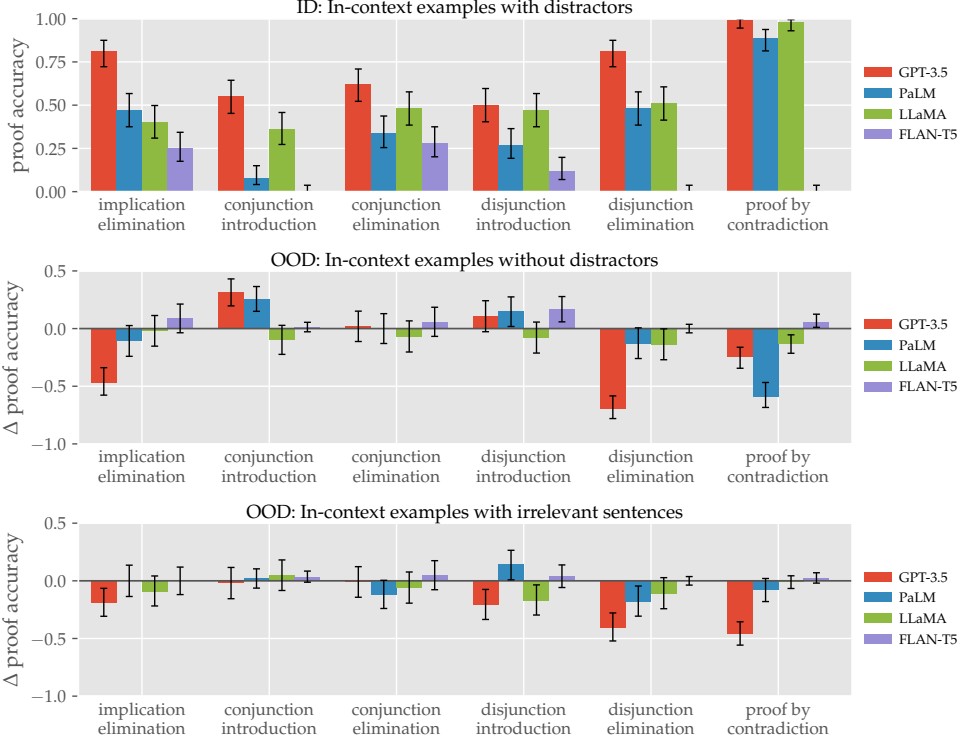

FIGURE 8: **(top)** Proof accuracy on examples where both the in-context examples and test examples have distractors. Sentences in all questions are ordered randomly. **(middle)** Proof accuracy when distractors are removed from the in-context examples, but not from the test examples. The sentences of the in-context questions have a fixed order (corresponding to a postorder traversal of the ontology tree), whereas the sentences in the test question have random order. See Figure 13 in the Appendix for the equivalent plot with absolute proof accuracy on the y-axis. **(bottom)** The same setting as (middle) except the distractors in the in-context examples are instead replaced with *irrelevant* sentences. Implication elimination examples have proof width of 1 and depth of 2. Conjunction introduction, conjunction elimination, and disjunction introduction examples have proof width 2 and depth 2. Disjunction elimination examples have proof width 3 and depth 1. Proof by contradiction examples have proof width 2 and depth 1.

> *Prove:*  Sally is dull.
>
> *Predicted answer:*  Sally is an impus.  Impuses are sterpuses.  Every grimpus is a zumpus.  Every brimpus is a grimpus.  Lorpuses are dumpuses.  Brimpuses are vumpuses. Every lorpus is a brimpus.  Sally is a brimpus.  Each vumpus is opaque.  Each dumpus is not brown.  Every grimpus is dull.  Sally is a grimpus.  Sally is dull.
>
> *Expected answer:*  Sally is a brimpus.  Every brimpus is a grimpus.  Sally is a grimpus.  Each grimpus is dull.  Sally is dull.

FIGURE 9: Example of an incorrect proof generated by GPT-3.5 on an OOD implication elimination example where the in-context demonstrations have no distractors, but the test example does. The premises (axioms) are given in blue, and invalid steps are given in red. For the full example, see Figure 16 in the Appendix.

visible in Figure 8. The models' performance is largely unaffected, with the exception of a few rules for specific models. This is in stark contrast to supervised learning, where it is always best to train on examples from the same distribution as the test example.

## 5   Conclusion and future work

In this study, we provide a systematic test of the general deductive reasoning capabilities of LLMs, specifically measuring their rule-, depth-, width-, and compositional generalization abilities. We found that LLMs exhibit mixed generalization to unseen deduction rules, but they exhibit more robust generalization to compositional proofs than previously suggested.

One important future direction is to better understand the mechanism of ICL and CoT prompting. We found that in many cases, for a given test example, the best in-context examples were drawn from a distribution distinct from that of the test example. This is not explained by existing theories of Bayesian inference [Xie et al., 2022] or gradient descent [Akyürek et al., 2023, Dai et al., 2023, von Oswald et al., 2022]. Are simpler examples better even if the test example is fairly complex? Should we include examples with a diverse set of deduction rules [Levy et al., 2022]? Or should the in-context examples focus on rules for which the model's OOD generalization is poor? Further study is needed to better characterize generalization from in-context examples.

## Reproducibility statement

For the sake of reproducibility of the analysis, model outputs (except those of PaLM), code for data generation, and analysis code are freely available with a permissive open-source license at github.com/asaparov/prontoqa. The generated data for all experiments in this paper is available in the file generated_ood_data.zip. The command python make_plots.py produces all figures used in this paper. GPT-3.5 experiments were run using the OpenAI API with the model text-davinci-003 on April $20^{th}$–$23^{rd}$, 2023. Experiments for Figure 7 and the bottom row of Figure 8 were run on August $3^{rd}$–$7^{th}$, 2023.

## Acknowledgements

We thank Tania Bedrax-Weiss, Xin Xu, and Deepak Ramachandran for their valuable feedback. This work was supported by Open Philanthropy, AWS AI, Samsung Advanced Institute of Technology (under the project Next Generation Deep Learning: From Pattern Recognition to AI), and in part through the NYU IT High Performance Computing resources, services, and staff expertise. NJ is supported by an NSF Graduate Research Fellowship under grant number 1839302.

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

# A Appendix

## A.1 Example from PRONTOQA-OOD

Q: Everything that is a lorpus, a brimpus, or a jompus is a shumpus. Every wumpus is a vumpus and a sterpus and a brimpus. Everything that is a vumpus, a grimpus, or a brimpus is a lempus. Everything that is a lempus or a jompus or a lorpus is a dumpus. Vumpuses are rompuses. Every sterpus is a gorpus. Everything that is a vumpus, a grimpus, or a brimpus is a dumpus. Wumpuses are shumpuses. Polly is a rompus. Polly is a wumpus.
Prove: Polly is a lempus or an impus or a yumpus.

A: Polly is a wumpus. Every wumpus is a vumpus and a sterpus and a brimpus. Polly is a vumpus and a sterpus and a brimpus. Polly is a brimpus. Polly is a vumpus, a grimpus, or a brimpus. Everything that is a vumpus, a grimpus, or a brimpus is a lempus. Polly is a lempus. Polly is a lempus, an impus, or a yumpus.

FIGURE 10: An example from PRONTOQA-OOD. This is a compositional example with a min depth of 4 and 3 rule types. The given answer is the expected chain-of-thought. The question is shown in blue, the query in red, and the chain-of-thought/answer in green.

## A.2 Results with absolute accuracy

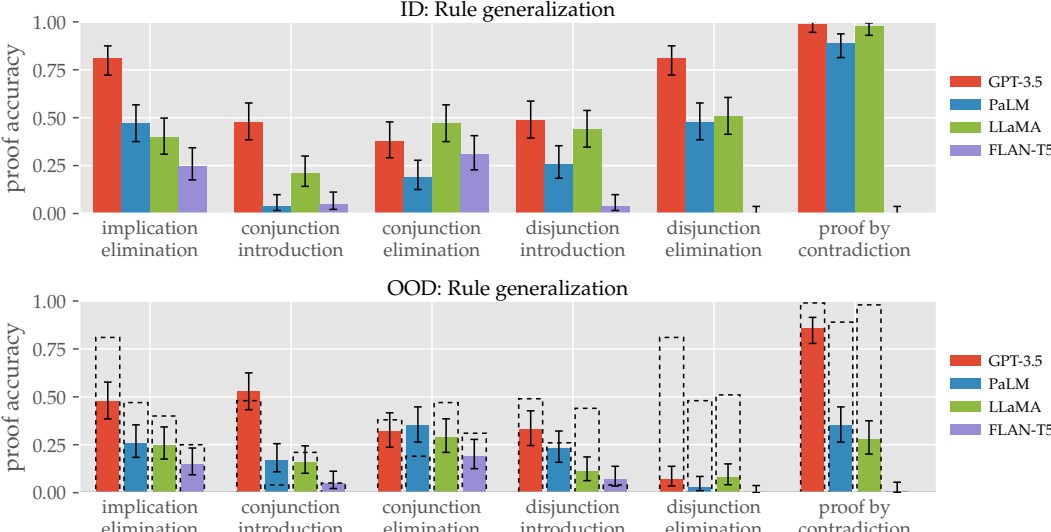

FIGURE 11: **(top)** Proof accuracy across examples with different deduction rules. The in-context examples and test examples come from the same distribution. **(bottom)** Proof accuracy where the test example is out-of-demonstration with respect to the in-context examples (for comparison, the in-demonstration proof accuracy is shown as the dotted black bars). That is, the test example has the specified deduction rule, but the in-context examples are uniformly distributed over *all other* deduction rules. See Figure 5 for an incorrect example. Implication elimination examples have proof width of 1 and depth of 2. Conjunction introduction, conjunction elimination, and disjunction introduction examples have proof width 3 and depth 2. Disjunction elimination examples have proof width 3 and depth 1. Proof by contradiction examples have proof width 2 and depth 1.

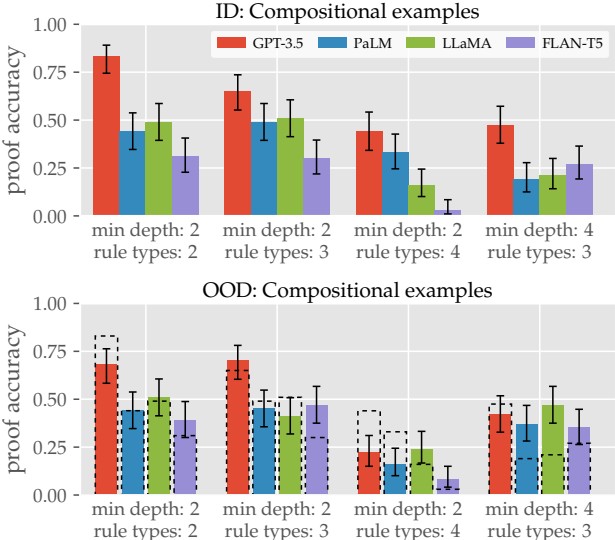

FIGURE 12: **(top)** Proof accuracy on compositional examples where the in-context examples are also compositional examples with the same min depth and number of rule types. **(bottom)** Proof accuracy where the test examples are compositional but the in-context examples are those of individual deduction rules (for comparison, the in-demonstration proof accuracy is shown as the dotted black bars).

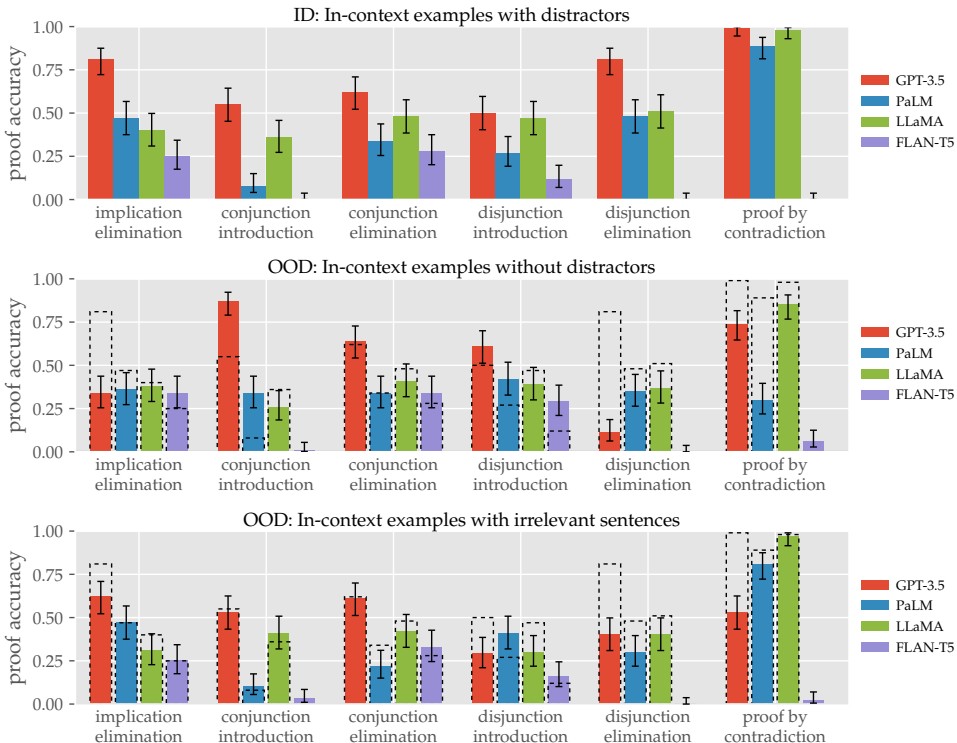

FIGURE 13: **(top)** Proof accuracy on examples where both the in-context examples and test examples have distractors. Sentences in all questions are ordered randomly. **(middle)** Proof accuracy when distractors are removed from the in-context examples, but not from the test examples (for comparison, the in-demonstration proof accuracy is shown as the dotted black bars). The sentences of the in-context questions have a fixed order (corresponding to a postorder traversal of the ontology tree), whereas the sentences in the test question have random order. **(bottom)** The same setting as (middle) except the distractors in the in-context examples are instead replaced with *irrelevant* sentences. Implication elimination examples have proof width of 1 and depth of 2. Conjunction introduction, conjunction elimination, and disjunction introduction examples have proof width 2 and depth 2. Disjunction elimination examples have proof width 3 and depth 1. Proof by contradiction examples have proof width 2 and depth 1.

## A.3 Full examples of incorrect predicted proofs

*Question:* Lempuses are gorpuses. Rompuses are wumpuses. Tumpuses are wumpuses. Dumpuses are gorpuses. Every zumpus is a wumpus. Every brimpus is a gorpus. Yumpuses are gorpuses. Lempuses are wumpuses. Rompuses are gorpuses. Tumpuses are gorpuses. Max is a tumpus or a rompus or a lempus. Max is a zumpus, a tumpus, or a rompus.

*Prove:* Max is a gorpus.

*Predicted answer:* Max is a tumpus or a rompus or a lempus. Max is a tumpus. Tumpuses are wumpuses. Max is a wumpus. Rompuses are gorpuses. Max is a gorpus. Max is a gorpus.

*Expected answer:* Assume Max is a tumpus. Tumpuses are gorpuses. Max is a gorpus. Assume Max is a rompus. Rompuses are gorpuses. Max is a gorpus. Assume Max is a lempus. Lempuses are gorpuses. Max is a gorpus. Since Max is a tumpus or a rompus or a lempus, Max is a gorpus.

FIGURE 14: Example of an incorrect proof generated by GPT-3.5 on an out-of-demonstration disjunction elimination example. The premises (axioms) are given in blue, and invalid steps are given in red.

*Question:* Polly is a wumpus. Every rompus is a dumpus. Lempuses are vumpuses. Everything that is a wumpus, a jompus, and a tumpus is a dumpus. Polly is a gorpus. Every impus is a brimpus. Everything that is a wumpus, a gorpus, and a shumpus is a vumpus. Each lempus is an impus and a lorpus and a rompus. Everything that is a wumpus, a jompus, and a tumpus is not a lorpus. Zumpuses are lorpuses. Polly is a shumpus. Polly is a zumpus. Polly is a zumpus. Polly is a wumpus. Polly is a jompus. Polly is a tumpus.

*Prove:* Polly is not a lempus.

*Predicted answer:* Polly is a wumpus, a jompus, and a tumpus. Everything that is a wumpus, a jompus, and a tumpus is not a lempus. Polly is not a lempus.

*Expected answer:* Polly is a tumpus. Polly is a jompus. Polly is a wumpus. Polly is a wumpus and a jompus and a tumpus. Everything that is a wumpus, a jompus, and a tumpus is not a lorpus. Polly is not a lorpus. Assume Polly is a lempus. Each lempus is an impus and a lorpus and a rompus. Polly is an impus and a lorpus and a rompus. Polly is a lorpus. This contradicts with Polly is not a lorpus. Polly is not a lempus.

FIGURE 15: Example of an incorrect proof generated by GPT-3.5 on an out-of-demonstration example with min depth 2 and 4 rule types. The premises (axioms) are given in blue, and invalid steps are given in red.

*Question:* Every grimpus is a zumpus. Each vumpus is opaque. Each dumpus is not brown. Each grimpus is dull. Lorpuses are moderate. Every brimpus is rainy. Impuses are temperate. Lorpuses are dumpuses. Brimpuses are vumpuses. Every lorpus is a brimpus. Zumpuses are angry. Grimpuses are yumpuses. Every brimpus is a grimpus. Impuses are sterpuses. Sally is an impus. Sally is a brimpus.

*Prove:* Sally is dull.

*Predicted answer:* Sally is an impus. Impuses are sterpuses. Every grimpus is a zumpus. Every brimpus is a grimpus. Lorpuses are dumpuses. Brimpuses are vumpuses. Every lorpus is a brimpus. Sally is a brimpus. Each vumpus is opaque. Each dumpus is not brown. Every grimpus is dull. Sally is a grimpus. Sally is dull.

*Expected answer:* Sally is a brimpus. Every brimpus is a grimpus. Sally is a grimpus. Each grimpus is dull. Sally is dull.

FIGURE 16: Example of an incorrect proof generated by GPT-3.5 on an OOD implication elimination example where the in-context demonstrations have no distractors, but the test example does. The premises (axioms) are given in blue, and invalid steps are given in red.

## A.4 Generative process details

In this section, we describe the process to generate examples of each deduction rule.

***Implication elimination*** (i.e. *modus ponens*)  Given $f(c)$ and $\forall x(f(x) \rightarrow g(x))$, prove $g(c)$. These are the examples in the original PRONTOQA. We follow the same process here:

1. Generate an ontology. For simplicity, we generate linear ontologies, consisting of a collection of concepts, as well as subtype-supertype relations between those concepts (i.e. concept $f$ is a subtype of the supertype $g$ if every instance of $f$ is an instance of $g$). For simplicity, we limit each type to have at most one supertype.
2. Perform a random walk of length $k$ from a randomly selected start vertex, where $k$ is the desired proof depth.
3. Traverse the edges of the ontology and convert each into a sentence of the question.
4. Convert each step of the random walk into a sentence of the gold chain-of-thought.

Note that this process allows us to generate proofs of any depth, but the width is fixed to $1$.

***Conjunction introduction***  Given $A$ and $B$, prove $A \wedge B$. The generative process is a modified version of that for implication elimination. Instead of generating rules of the form $\forall x(f(x) \rightarrow g(x))$, we generate rules of the form $\forall x(f_1(x) \wedge \ldots \wedge f_n(x) \rightarrow g(x))$, where $n$ is the proof width. Given, $f_1(c)$, …, and $f_n(c)$, the model must first prove $f_1(c) \wedge \ldots \wedge f_n(c)$ before applying implication elimination to prove $g(c)$. To increase the depth of the proof, $g(c)$ itself can be part of a conjunct in the antecedent of another rule.

***Conjunction elimination***  Given $A \wedge B$, prove $A$. These examples are identical to those in conjunction introduction, except the conjunction appears in the consequent of each rule, rather than in the antecedent: $\forall x(f(x) \rightarrow g_1(x) \wedge \ldots \wedge g_n(x))$ where $n$ is the proof width.

***Disjunction introduction***  Given $A$, prove $A \vee B$. These examples are identical to those in conjunction introduction, except the conjunction is replaced with disjunction: $\forall x(f_1(x) \vee \ldots \vee f_n(x) \rightarrow g(x))$ where $n$ is the proof width. But note that grounded axioms are not necessary for every disjunct: To apply the rule $\forall x(f_1(x) \vee \ldots \vee f_n(x) \rightarrow g(x))$, knowing $f_n(c)$ is sufficient, and we do not need to generate grounded axioms for the other disjuncts $f_i(c)$ for $i < n$.

***Disjunction elimination*** (i.e. *proof by cases*)  Given $A_1 \vee \ldots \vee A_n$, and $A_i \vdash C$ for all $i$, prove $C$. Here, $n$ is the proof width. While it is possible to construct proofs containing multiple nested applications of disjunction elimination, such proofs are quite complex, even for humans to understand, and so we fix the depth of these examples to $1$. To generate an example, we first generate the disjunction: $f_1(c) \vee \ldots \vee f_n(c)$. Next, generate the rules for each case: $\forall x(f_i(x) \rightarrow g(x))$ for all $i$. The goal is to prove $g(c)$.

***Proof by contradiction***  Given $A \vdash B$ and $\neg B$, prove $\neg A$. Note that this is a rule composed of two natural deduction rules: negation elimination and introduction. But since those individual rules do not lend themselves to a natural text representation, we choose to study their composition. Similar to disjunction elimination, it is possible to construct proofs containing multiple nested applications of proof by contradiction, but such proofs are unnaturally complex. So we fix the depth to $1$. To generate an example, we first generate an axiom $\neg g(c)$. Next, for each subproof, we generate a rule $\forall x(f_1(x) \vee \ldots \vee f_n(x) \rightarrow g(x))$, where $n$ is the proof width. The goal is to prove $\neg f_1(c) \wedge \ldots \wedge \neg f_n(c)$. Note that in addition to proof by contradiction, this proof requires disjunction introduction, implication elimination, and conjunction introduction.

Note that the above list constitutes a complete set of deduction rules from propositional natural deduction, save for one rule: implication introduction. However, it is unclear how to construct an example with this deduction rule where its difficulty can be controlled by increasing the width or depth of the proof (e.g. how can a statement of the form $A_1 \rightarrow A_2 \rightarrow \ldots \rightarrow A_n$ be expressed in natural language?).

**Algorithm 1:** Pseudocode for generating examples of compositional proofs in PRONTOQA-OOD. In this algorithm, $\Omega$ denotes the set of all logical forms. The function `generate_compositional_proof` is initially called with parameters $\Omega$, $\varnothing$, $d$, $e$, and `false`, where $d$ is the requested depth and $e$ is a randomly selected entity name (e.g. `alex`, `fae`, etc). `sample` is a helper function that, given an input set of logical forms $S$ and an entity $e$, returns `sample_uniform`({set of logical forms in $S$ with minimal depth where all atoms are of the form $t(e)$ where $t$ is a predicate}).

---

1   **function** `generate_compositional_proof` (*set of possible conclusions (logical forms) $C$,*
                                          *disallowed deduction rules $R$,*
                                          *requested depth d,*
                                          *ground entity e,*
                                          *is proof hypothetical h*)

2   initialize $A$ as the set of all deduction rules excluding those in $R$
     /* filter deduction rules such that: (1) an element of $C$ can be a conclusion
        of the rule, (2) for which we have sufficient depth, and (3) we don't create
        overly complex logical forms */
3   **if** *$C$ does not contain a conjunction*
4      set $A = A \setminus \{$`conjunction_introduction`$\}$
5   **if** *$C$ does not contain a disjunction*
6      set $A = A \setminus \{$`disjunction_introduction`$\}$
7   **if** *$h = $ true or $d = 1$ or $C$ does not contain a negation*
8      set $A = A \setminus \{$`proof_by_contradiction`$\}$
9   **if** *$h = $ true or $d = 1$ or $C$ contains only conjunctions or only disjunctions*
10  set $A = A \setminus \{$`disjunction_elimination`$\}$
11  **if** *$C$ contains only conjunctions or only disjunctions*
12     set $A = A \setminus \{$`conjunction_elimination`$\}$
13  **if** *$C$ contains only conjunctions or only disjunctions and any operand is negated*
14     set $A = A \setminus \{$`implication_elimination`$\}$
15  **if** *$d = 0$ or $C$ contains a singleton logical form or $A = \varnothing$*
16     **return** *axiom* step with conclusion given by *sample*$(C, e)$
17  $r = $ `sample_uniform`$(A)$
18  **if** *$r = $ implication_elimination*
19     **do**
20        for any $c \in C$, $a$ and $c$ share any operands or negations of operands
21     **while** $a = $ `generate_compositional_proof`$(\Omega, \varnothing, d - 1, e, h)$
22     **do**
23        $a$ and $s$ do not share any operands or negations of operands
24     **while** $s = $ *sample*$(C, e)$
25     **return** *implication_elimination* with premises $a$ and $\forall x(a[e \rightarrow x] \rightarrow s[e \rightarrow x])$
26  **else if** *$r = $ conjunction_introduction*
27     initialize $P$ as an empty list, and $i = 0$
28     $L = |C|$ if $C$ contains only conjunctions, else $L = 3$
29     **do**
30        let $C_i = i^{th}$ operand of $C$ if $C$ contains only conjunctions, else $C_i = \Omega$
31        $a = $ `generate_compositional_proof`$(C_i, \{$`conjunction_elimination`$\}, d - 1, e, h)$
32        **if** *$a$ is atomic and $a$ is not any other operand of $C$*
33           append $a$ to $P$
34           $i = i + 1$
35     **while** $i < L$
36     **return** *conjunction_introduction* with premises $P$
37  **else if** *$r = $ conjunction_elimination*
38     let $C'$ be the set of conjunctions of length 3, $i = $ `sample_uniform`$(\{1, 2, 3\})$
39     $C' = \{c \in C' : $ the $i^{th}$ operand of $c'$ is in $C\}$
40     **do**
41        $a = $ `generate_compositional_proof`$(C', \{$`conjunction_introduction`$\}, d - 1, e, h)$
42     **while** *$a$ has no duplicate operands, and each operand of $a$ is not itself a conjunction or disjunction*
43     **return** *conjunction_elimination* with premise $a$ and conclusion given by the $i^{th}$ operand of $a$

**Algorithm 1:** (continued from previous page)

```
44  else if r = disjunction_introduction
45    if C = Ω  let C be the set of disjunctions of length 3
46    i = sample_uniform( number of disjuncts in C)
47    do
48      let C_i = i^th operand of C
49      a = generate_compositional_proof(C_i, {disjunction_elimination}, d − 1, e, h)
50    while a is atomic and a is not any other operand of C
51    replace i^th operand of C with a
52    do
53      x = sample(C, e)
54    while i^th disjunct of x is distinct from all other disjuncts
55    return disjunction_introduction with premise given by the i^th operand of x and conclusion x
56  else if r = disjunction_elimination
57    initialize P as an empty list
58    while |P| < 2 do
59      p = generate_compositional_proof(C, {disjunction_introduction}, d − 1, e, true)
60      if p is not a conjunction or disjunction and p has an axiom that is not an axiom of any q ∈ P
61        append p to P
62    let A_i be the set of axioms of P_i that are not axioms of P_j for i ≠ j
63    let a_i = sample_uniform(A_i) for all i
64    let a' be a disjunction with disjuncts a_i
65    a = generate_compositional_proof({a'}, {disjunction_introduction}, d − 1, e, h)
66    return disjunction_introduction with premises a and P_i
67  else if r = proof_by_contradiction
68    let N be the set of all negated logical forms
69    a = generate_compositional_proof(N, {proof_by_contradiction}, d − 1, h)
70    do
71      let a = ¬s
72      b = generate_compositional_proof({s}, {proof_by_contradiction}, d − 1, e, true)
73    while b has an atomic non-negated axiom that is not an axiom of a
74    s' = sample_uniform({atomic non-negated axioms of b that are not axioms of a})
75    return proof_by_contradiction with premises a and b and conclusion ¬s'
```

## A.5 Generating compositional proofs

We use a simple recursive procedure to generate compositional proofs: (1) select a deduction rule uniformly at random, (2) select the premises for the selected rule, (3) recursively generate a subproof for each premise. A consistency checking step is required to make sure we avoid generating contradictory axioms.[6] In addition, we avoid generating an elimination rule directly following an introduction rule (or vice versa).[7] See Algorithm 1 in the appendix for pseudocode of this procedure. To test compositional proofs of various sizes, we implement a parameter that controls the minimum depth of the proof tree, and another parameter that controls the number of distinct rule types in each proof.

## A.6 Further details on evaluation of CoT

We aim to test whether LLMs are able to use deduction rules OOD, where the rules do not appear in the in-context examples, and we take care not to be overly strict. For example, we wish to avoid penalizing the model for formatting differences, so long as the reasoning is correct. To this end, in determining whether a logical form follows from previous logical forms, we consider any deduction rule listed in Table 2. We also allow for two additional rules: (1) given $\forall x(f(x) \to g(x))$ and

---

[6]An example is: Suppose we select conjunction introduction as the first rule; next, we recursively generate the proof of each conjunct; suppose for each of these, we choose to generate the axioms `cat(alex)` and `¬cat(alex)`.

[7]In the following example, a conjunction introduction step immediately follows a conjunction elimination step: "Jay is a cat and orange. Jay is a cat. Jay is orange. Jay is a cat and orange."

$\forall x(g(x) \to h(x))$ conclude $\forall x(f(x) \to h(x))$, and (2) given $\forall x(f(x) \to g(x))$ and $\neg g(c)$ conclude $\neg f(c)$ (i.e. modus tollens).[8] Additionally, we are flexible with respect to the ordering of conjuncts and disjuncts. For example, given the previous steps $f(a) \wedge g(a)$ and $\forall x(g(x) \wedge f(x) \to u(x) \vee v(x))$, we consider $v(a) \vee u(a)$ to be valid.

## A.7 Generating distractors

***Implication elimination*** For any rule $\forall x(f(x) \to g(x))$ in the gold proof, we generate a distractor rule $\forall x(f(x) \to h(x))$ where the concept $h$ is a distractor and is not helpful in completing the proof. In addition, for any ground logical form in the gold proof $f(c)$, we generate a distractor logical form $h(c)$ as well as a rule $\forall x(h(x) \to h'(x))$. Note that the original PRONTOQA only adds a single distractor, whereas we add multiple, one for each hop in the proof.

***Conjunction introduction*** Similar to those in implication elimination. For any rule $\forall x(f_1(x) \wedge \ldots \wedge f_n(x) \to g(x))$, we generate a rule of the form $\forall x(h_1(x) \wedge \ldots \wedge h_{n-1}(x) \wedge f_n(x) \to g(x))$ where $h_i$ are distractor concepts. Grounded distractor conjuncts are also generated as axioms $h_i(c)$, so that, given $f_n(c)$, both the gold rule and distractor rule are valid proof steps.

***Conjunction elimination*** Distractors are generated similarly to the conjunction introduction case.

***Disjunction introduction*** Distractors are generated similarly to the conjunction introduction case.

***Disjunction elimination*** Since this deduction step has many premises, multiple distractors are necessary to ensure the model doesn't resort to heuristics. For every rule of the form $\forall x(f_i(x) \to g(x))$, two distractor rules are generated: $\forall x(f_i(x) \to h'(x))$ and $\forall x(h_i(x) \to g(x))$. A distractor disjunction is also generated: $h''(c) \vee h_1(c) \vee \ldots \vee h_{n-1}(c)$.

***Proof by contradiction*** As with disjunction elimination, multiple distractors are necessary here. We generate two distractor rules $\forall x(f_1(x) \vee \ldots \vee f_n(x) \to h(x))$ and $\forall x(h_1(x) \vee \ldots \vee h_n(x) \to g(x))$. We also generate the distractor axiom $\neg h'(c)$ so that the model is forced to choose between two axioms for the first step of the proof.

To avoid creating inconsistencies when generating a distractor rule, we avoid using existing predicates in the consequent of each rule.

---

[8]Analogous to the *broadly-valid* steps in PRONTOQA.

**Algorithm 2:** Pseudocode for evaluating the output chain-of-thought. Here, the comparison operations between logical forms ignore the order of conjuncts if both operands are conjunctions; and similarly for disjunctions. In addition, when iterating over previous steps in the proof, we consider them in reverse order, so that more recent steps are prioritized. The helper function `negate` is defined, in order of precedence: $\texttt{negate}(\neg A) = A$, $\texttt{negate}(A \vee B) = \texttt{negate}(A) \wedge \texttt{negate}(B)$, $\texttt{negate}(A \wedge B) = \texttt{negate}(A) \vee \texttt{negate}(B)$, or $\texttt{negate}(A) = \neg A$.

```
1  function evaluate_cot(context sentences Q₁,…,Qₘ,
                         predicted chain-of-thought sentences C₁,…,Cₙ,
                         goal sentence g)
```
2    $L^g = \texttt{semantic\_parse}(g)$                                           `/* parse the goal */`

3    **for** $i \in 1,\ldots,m$ **do**                                             `/* parse the context */`

4      $L_i^Q = \texttt{semantic\_parse}(Q_i)$

5    **for** $i \in 1,\ldots,m$ **do**                   `/* parse the predicted chain-of-thought */`

6      $L_i^C = \texttt{semantic\_parse}(C_i)$

7    initialize $S$ as an empty set, and $H$ as an empty map

8    **for** $i \in 1,\ldots,n$ **do**             `/* reconstruct the proof from the chain-of-thought */`

9      **if** $L_i^C$ indicates 'this is a contradiction'

10        **if** $\texttt{negate}(L_{i+1}^C) \in H(L_{i-1}^C)$

11          $(P, D, k) = (\{L_{i-1}^C, \texttt{negate}(L_{i+1}^C)\}, \{\texttt{negate}(L_{i+1}^C)\}, 1)$

12        **else continue**

13      **else**

14        $(P, D, k) = \texttt{is\_provable}(L_i^C, \{L_1^Q, \ldots, L_m^Q\}, S, H)$

15      set $H(L_i^C) = \bigcup_{p \in P} H(p) \setminus D$

16      **if** $k \geq 0$

17        add $L_i^C$ to $S$

18    **return** $L^g \in S$       `/* the proof is correct if the final conclusion is provable */`

19 **function** `is_provable`(*logical form* $\varphi$,
                              *set of axioms* $A$,
                              *previous conclusions* $S$,
                              *hypothesis map* $H$)

20    **if** $\varphi \in A$

21      **return** $(\{\varphi\}, 1)$                                  `/* this is an axiom */`

22    **else if** $\varphi$ is a conjunction or disjunction

23      initialize $P'$ as an empty list, and $k' = 0$

24      **for** $\varphi_i$ operand in $\varphi$ **do**

25        $(P, k) = \texttt{is\_provable}(\varphi_i, A, S, H)$

26        **if** $\varphi$ is a conjunction

27          **if** $k \geq 0$ and the step immediately preceding $\varphi$ in the proof is in $P$

28            append $P$ to $P'$

29            set $k' = k' + k$

30          **else break**

31        **else if** $k > 0$ and $\varphi$ is a disjunction

32          **return** $(P, \varnothing, k + 1)$          `/* provable by disjunction introduction */`

33      **if** $P'$ has the same size as $\varphi$ has operands

34        **return** $(\bigcup P', \varnothing, k')$          `/* provable by conjunction introduction */`

35    **for** $a \in S \cup A$ **do**

36      **if** $a$ is a conjunction and $\varphi = a_i$ for some $i$

37        **return** $(\{a\}, \varnothing, 1 + \mathbb{1}\{a \in A\})$       `/* provable by conjunction elimination */`

38      **else if** $a$ has form $\forall x(\psi \to \gamma)$ where $\gamma[x \mapsto c] = \varphi$

39        $(P, k) = \texttt{is\_provable}(\psi[x \mapsto c], A, S, H)$

40        **if** $k \geq 0$ and the step immediately preceding $\varphi$ in the proof is in $P \cup \{a\}$

41          **return** $(P \cup \{a\}, \varnothing, k + \mathbb{1}\{a \in A\})$     `/* provable by conjunction elimination */`

42    **for** $s \in S$ where $s$ is a disjunction **do**

43      **if** for all disjuncts $s_i$, there is a $s_j \in S$ such that $s_j = \varphi$ and $s_i \in H(s_j)$

44        **return** $(\{s_j\}, \{s_i\}, 1)$            `/* provable by disjunction elimination */`

**Algorithm 2:** (continued from previous page)

```
45   for a ∈ S ∪ A do
46       if a has form ∀x(ψ → γ) where γ[x ↦ c] = φ
47           (P, k) = is_provable(ψ[x ↦ c], A, S, H)
48           if k ≥ 0 and the step immediately preceding φ in the proof is in P ∪ {a}
49               return (P ∪ {a}, ∅, k + 1{a ∈ A})        /* provable by implication elimination */

50       else if a has form ∀x(ψ → γ) where negate(ψ[x ↦ c]) = φ
51           (P, k) = is_provable(negate(γ[x ↦ c]), A, S, H)
52           if k ≥ 0 and the step immediately preceding φ in the proof is in P ∪ {a}
                 /* provable with additional deduction rules (modus tollens) */
53               return (P ∪ {a}, ∅, k + 1{a ∈ A})

54   if φ ∈ S
55       return ({φ}, ∅, 0)                               /* proved by previous step */
56   else if φ has form ∀x(ψ → γ)
         /* note:  we precompute this graph */
57       let G be the graph where for any axiom in A with form ∀x(α → β), α and β are vertices and there is a
           directed edge from α to β
58       if there is a path in G from ψ to γ
             /* provable with additional deduction rules */
59           return ( axioms corresponding to path edges , ∅, length of path )

60   return (∅, ∅, −1)                                    /* this step is not provable (i.e., invalid) */
```

