# OpenReview forum: "Testing the General Deductive Reasoning Capacity of Large Language Models Using OOD Examples"
_NeurIPS.cc/2023/Conference — NeurIPS 2023 poster_

### Official Review · Reviewer_bcxG · 2023-06-13

**Soundness:** 3 good
**Presentation:** 2 fair
**Contribution:** 2 fair
**Rating:** 6
**Confidence:** 5

**Summary:**

The paper presents a programmable natural language reasoning dataset, PrOntoQA-OOD, specifically designed for testing the out-of-distribution generalization abilities of machine learning models. 4 LLMs, FLAN-T5, LLaMA, GPT-3.5, and PaLM, are tested on this dataset for their ability to generalize to different deduction rules, proof width/depth, and rule compositions.

**Strengths:**

- The dataset construction is solid and takes into consideration various deduction rules.
- The experimentation is thorough and insightful.
- The paper is well written.

**Weaknesses:**

- (Major) [1] presents a benchmark for theorem proving that is programmable and tests generalization across six dimensions. These dimensions include different deduction rules and proof width/depth, which this paper also discusses. It's a very relevant paper and the failure to discuss it as related work in context seems like a major oversight in scholarship. But this is quite easy to fix.
- (Minor) Footnote 2 says the deduction rules used in this paper have minor differences with Gentzen's natural deduction, yet there are 3 rules omitted from it. This seems more than minor and I fail to see how natural deduction's completeness or soundness claims can translate to this paper's, without substantiation.
- (Minor) Figure 3 notes GPT 3.5 as 175B* with an asterisk indicating low confidence in its size. I feel like it's best to omit the 175B size claim altogether if the authors are unsure, to avoid misleading.
- (Major) For the figures, OOD generalization performances are shown in $\Delta$ proof accuracies w.r.t. the ID ones. This is hard for me to get an intuitive understanding of the different models used, particularly for FLAN-T5: FLAN-T5 often has trivial (close to 0) proof accuracies for ID and its OOD performance $\Delta$ being 0 is an indication of triviality instead of good generalization. Showing the absolute OOD proof accuracies will solve this problem.
- (Minor) For Figure 5 and several others, there's an example of (predicted answer, expected answer), without the question. I had to use my inductive reasoning to figure out exactly what the question was. Please fix this.
- (Minor) Line 17 seems to cast theorem proving and medical diagnosis as purely deductive reasoning problems. Both tasks require inductive reasoning among other modes. So this claim is not rigorous.

Overall, there are problems but all seem fixable. I'm happy to change my rating if the authors can amend the paper to be more satisfactory.


[1] Wu, Y., Jiang, A.Q., Ba, J. and Grosse, R., 2020. Int: An inequality benchmark for evaluating generalization in theorem proving. arXiv preprint arXiv:2007.02924.

**Questions:**

More contextualization with prior work should be good.

**Limitations:**

More discussion on limitation of this paper's approach for evaluating LLMs reasoning would certainly improve the paper.

---

> ### Author Rebuttal · Authors · 2023-08-10
>
> Thank you for the feedback and helpful comments! We would greatly appreciate any additional comments or questions during the discussion period.
>
> 1. “[1] presents a benchmark for theorem proving that is programmable and tests generalization across six dimensions. These dimensions include different deduction rules and proof width/depth, which this paper also discusses.”
>
> We thank the reviewer for bringing this work to our attention. The referenced work focuses on symbolic theorem proving but otherwise shares many aspects of the exploration with our work. We will add it to the paragraph beginning on line 65.
>
> [1] Wu, Y., Jiang, A.Q., Ba, J. and Grosse, R., 2020. Int: An inequality benchmark for evaluating generalization in theorem proving. arXiv preprint arXiv:2007.02924.
>
>
> 2. “For the figures, OOD generalization performances are shown in proof accuracies w.r.t. the ID ones. This is hard for me to get an intuitive understanding of the different models used, particularly for FLAN-T5: FLAN-T5 often has trivial (close to 0) proof accuracies for ID and its OOD performance being 0 is an indication of triviality instead of good generalization. Showing the absolute OOD proof accuracies will solve this problem.”
>
> This is a good point. To help readers better interpret the OOD charts in relation to the ID charts, we will add the following sentence to line 168: “We emphasize that the \Delta proof accuracies in the bottom row of the figure should be interpreted in comparison with the accuracies in the top row (e.g. for some rules, the zero \Delta accuracy for FLAN-T5 is due to zero absolute accuracy).”
>
>   We tried different ways to plot the OOD accuracy on an absolute scale, showing the change in accuracy vs ID as well as 95% confidence intervals, but we found the resulting charts to be difficult to read. But we will provide OOD charts with absolute proof accuracies in the Appendix.
>
>
> 3. “Footnote 2 says the deduction rules used in this paper have minor differences with Gentzen's natural deduction, yet there are 3 rules omitted from it. This seems more than minor and I fail to see how natural deduction's completeness or soundness claims can translate to this paper's, without substantiation.”
>
> This is a fair point. We clarify that we do not make claims about the soundness and completeness of our modified proof calculus. Rather, we use the soundness and completeness of natural deduction (ND) to motivate its use in our experiments.
>
> But it is interesting to think about the soundness and completeness of the modified proof calculus. Since the proof by contradiction rule can be proved using negation elimination followed by negation introduction, any proof in our system can be translated into one in ND, which is sound. Therefore, our system is also sound. Our system is not complete, however, since it is missing rules governing truth and falsity. We chose to remove these rules since it is very non-obvious how to express the logical forms “T” or “F => cat(bob)” in natural language. The removed deduction rules also seem very unlikely to appear outside the context of mathematical proofs (and similar contexts). For logical forms without “T” or “F,” our system would be complete with respect to minimal logic, since minimal logic excludes the principle of explosion (i.e. falsity elimination).
>
> We would be happy to include this discussion in an appendix section if the reviewer finds it helpful.
>
>
> 4. “For Figure 5 and several others, there's an example of (predicted answer, expected answer), without the question. I had to use my inductive reasoning to figure out exactly what the question was. Please fix this.”
>
> We thank the reviewer for this suggestion. We chose not to include the full question due to space constraints: Each question not only includes the query, but all the axioms required to answer the question, as well as all distractors. However, we will modify the figures to include the query (e.g. “Prove: Polly is not a lempus.”) and we will add the full questions to the appendix, with references from each figure.
>
>
> 5. “Line 17 seems to cast theorem proving and medical diagnosis as purely deductive reasoning problems. Both tasks require inductive reasoning among other modes. So this claim is not rigorous.”
>
> We will rephrase the sentence on line 17 as: “In many tasks that require deductive reasoning, such as theorem proving or medical diagnosis, the complexity of proofs can grow without bound via the use of multiple deduction rules and the composition of subproofs.”

---

> > ### Comment · Reviewer_bcxG · 2023-08-13
> >
> > Thank you for the response. I am happy that the suggestions are well received and the responses are satisfactory. Given that the major issues (contextualization and chart visibility) are fixed in the camera-ready, I think the paper warrants acceptance.
> >
> > I have changed the rating to reflect this new position.

---

### Official Review · Reviewer_CjCV · 2023-07-05

**Soundness:** 3 good
**Presentation:** 3 good
**Contribution:** 3 good
**Rating:** 7
**Confidence:** 4

**Summary:**

This submission proposes a new dataset PRONTOQA-OOD to systematically study the general deductive reasoning capabilities of large language models (LLMs). Some findings are obtained, e.g., LLMs exhibit mixed generalization to unseen deduction rules and more robust generalization to longer, wider, and compositional proofs than previously suggested.

**Strengths:**

Summary:
(1) A new dataset for evaluating the deductive reasoning capabilities of LLMs is proposed.
(2) Some new findings on the deductive reasoning capabilities of LLMs are presented.
(3) The submission is well written and easy to follow.

Soundness:
The construction method for the proposed dataset PRONTOQA-OOD is sound, using natural language templates from deduction rules and fictional names for all concepts, where fictional names rather than real names are exploited to prevent LLMs from reusing knowledge from pretraining. The findings are drawn from careful experimental designs and adequately supported with empirical evidence.

Presentation:
The submission is well written and organized, with sufficient technical details, easy to follow.

Contribution:
Evaluating the deductive reasoning capabilities of LLMs is important in the field of artificial general intelligence. The submission pushes forward this study by considering more natural deduction rules and drawing some new findings. The findings are valuable to share with the broader NuerIPS community.


**Weaknesses:**

Similar to existing datasets for evaluating the deductive reasoning capabilities of LLMs, the proposed dataset is built on simple and short templates mimicking deduction rules. The sentences used to evaluate LLMs are different from general natural language sentences. It may not reflect the deductive reasoning capabilities of LLMs over natural language sentences.

There are two minor issues on presentation. One issue is on the definition of the same distribution, which has been referred to in several places including Abstract. It is unknown how to determine whether two examples are obtained from the same distribution or from different distributions. The other issue is on the mentions of abbreviations. An abbreviation should be given its full name when it is mentioned for the first time, however, some abbreviations such as OOD do not come with their full names when they appear for the first time.


**Questions:**

How is the same distribution defined? That is, how do we determine whether two examples are obtained from the same distribution?

**Limitations:**

I do not find limitations that the authors do not address in their methodology or mention in future work.

---

> ### Author Rebuttal · Authors · 2023-08-10
>
> Thank you for the feedback and helpful comments! We would greatly appreciate any additional comments or questions during the discussion period.
>
> 1. “Similar to existing datasets for evaluating the deductive reasoning capabilities of LLMs, the proposed dataset is built on simple and short templates mimicking deduction rules. The sentences used to evaluate LLMs are different from general natural language sentences. It may not reflect the deductive reasoning capabilities of LLMs over natural language sentences.”
>
> The synthetic aspect of our dataset and the original PrOntoQA facilitated a much greater degree of programmability and control over many variables that can have a large effect on the model’s reasoning performance. For example, using real-world sentences would confound the model’s ability to memorize and recall information from its pretraining with its ability to reason [1]. This was shown in [2] where the model’s reasoning accuracy was significantly lower when real concept names were replaced with fictional ones. We also note that recent LMs are quite good at understanding realistic examples, as evidenced by recent work on QA and math word problems, and we don’t believe that using examples with more naturalistic language will affect our main observations.
>
> [1] Zhaofeng Wu, Linlu Qiu, Alexis Ross, Ekin Akyürek, Boyuan Chen, Bailin Wang, Najoung Kim, Jacob Andreas, & Yoon Kim (2023). Reasoning or Reciting? Exploring the Capabilities and Limitations of Language Models Through Counterfactual Tasks. CoRR, abs/2307.02477. URL https://arxiv.org/abs/2307.02477.
>
> [2] Abulhair Saparov and He He. Language models are greedy reasoners: A systematic formal analysis of chain-of-thought. ICLR 2023. URL https://arxiv.org/abs/2210.01240.
>
>
> 2. “How is the same distribution defined? That is, how do we determine whether two examples are obtained from the same distribution?” “One issue is on the definition of the same distribution, which has been referred to in several places including Abstract. It is unknown how to determine whether two examples are obtained from the same distribution or from different distributions.”
>
> An advantage of a synthetic and programmable dataset such as PrOntoQA and PrOntoQA-OOD is that we define the generative process with which we generate the examples, and so we control the ground-truth distribution each example is sampled from, and thus are able to make claims such as whether two examples come from the same or different distributions.
>
>
> 3. “An abbreviation should be given its full name when it is mentioned for the first time, however, some abbreviations such as OOD do not come with their full names when they appear for the first time.”
>
> We thank the reviewer for the suggestion. We will rephrase “OOD” on line 41 to “out-of-demonstration (OOD).” We performed a search for other undefined acronyms and found no others.

---

### Official Review · Reviewer_ReKG · 2023-07-07

**Soundness:** 3 good
**Presentation:** 3 good
**Contribution:** 3 good
**Rating:** 7
**Confidence:** 3

**Summary:**

In this paper, the authors explore the general deductive reasoning ability of LLMs. They test LLMs on a broad set of deduction rules and measure the ability to generalize to more complex proofs from simpler demonstrations from multiple angles. They construct a programmable dataset that enables control over deduction rules and proof complexity. The experiments are conducted on four LLMs with different sizes and training objectives.

**Strengths:**

Overall, I think the paper is addressing an important research direction that helps better understand the general deductive reasoning ability of LLMs. Several strengths I particularly admire are:

1. The category of rules and studies of models under various settings are systematic and comprehensive. Many takeaways are provided (although I believe some of them can be made clearer).
2. The creation of a new synthetic and programmable reasoning dataset is a valuable contribution. This dataset allows for control over deduction rules and proof complexity, which can be beneficial for future research in this area.

**Weaknesses:**

While the paper presents comprehensive experiments to assess the reasoning capabilities of LLMs, there are aspects that require clarification:

1. The main issue with the writing is that the outcomes of the experiments are not clearly communicated, likely due to a lack of correlation between figures and claims. For example, the paper asserts that "LLMs can generalize to more complex proofs but require explicit demonstrations to produce hypothetical subproofs". It would be beneficial if the authors could explicitly highlight which experiments (and comparisons) substantiate this claim.

2. A critical concern, albeit not too severe, is the absence of a discussion about the underlying mechanisms and the implications or applications of the observations. Various claims are made, but the *reasons* behind these phenomena and *how* we can improve the models aren't elaborated on. For instance, ICL exhibits variable performance under different deduction rules, which complicates the decision of its usage. If the authors could provide explanations, identify the factors causing ICL's performance degradation, and propose potential solutions, it would significantly enhance the paper's value.

**Questions:**

1. In line 183, "we observe that the gap in proof accuracy between the ID and OOD settings is quite small". However, in figure 6, it seems like you observe an accuracy drop of 20% in some experiments.

2. Different from previous works, this paper claims that LLMs can generalize compositionally. Can you further compare your work with them, and analyze why different conclusions are made?

3. I cannot understand the subtitle of section 4.1. Many a more concise and academic title would be better.

**Limitations:**

Not applicable.

---

> ### Author Rebuttal · Authors · 2023-08-10
>
> Thank you for the feedback and helpful comments! We would greatly appreciate any additional comments or questions during the discussion period.
>
> 1. “The main issue with the writing is that the outcomes of the experiments are not clearly communicated, likely due to a lack of correlation between figures and claims. For example, the paper asserts that "LLMs can generalize to more complex proofs but require explicit demonstrations to produce hypothetical subproofs". It would be beneficial if the authors could explicitly highlight which experiments (and comparisons) substantiate this claim.”
>
> We thank the reviewer for the suggestion to make clearer the connections between claims and experiments. At the end of Section 1, in the bulleted list of main claims, we will add references to the subsections containing the experiments that substantiate the claim. More specifically:
>  - On line 41, we will add explicit references to Subsection 4.2.2 and Figure 6.
>  - On line 45, we will add explicit references to Subsections 4.2.2, 4.3, and Figures 6 and 9.
>  - On line 49, we will add explicit references to Subsection 4.2.1 and Figure 4.
>
> In addition, after each experiment described in the Results section, we will reiterate the main claims supported by that experiment.
>
> 2. “...the reasons behind these phenomena and how we can improve the models aren't elaborated on. For instance, ICL exhibits variable performance under different deduction rules, which complicates the decision of its usage.”
>
> We agree that the paper would benefit from additional elaboration and discussion on these points. We added further discussion as described in the common rebuttal. Specifically, we make a distinction between deduction rules with which the model is more vs less familiar. Our results show that in-demonstration examples are required for proof by cases and proof by contradiction, but not for other deduction rules, highlighting weaknesses where the models may benefit from explicit demonstrations, additional pretraining, and/or finetuning.
>
>
> 3. “In line 183, `we observe that the gap in proof accuracy between the ID and OOD settings is quite small’. However, in figure 6, it seems like you observe an accuracy drop of 20% in some experiments.”
>
> We agree this characterization could be more nuanced. We will rephrase the sentence on 183 to: “Figure 6, in all but three experiments, we observe that the gap in proof accuracy between the ID and OOD settings is close to zero…”
>
>   It is also important to interpret the OOD \Delta proof accuracies in comparison to the absolute proof accuracies shown in the ID charts. To underline this for the reader, we will add the following sentence to line 168: “We emphasize that the \Delta proof accuracies in the bottom row of the figure should be interpreted in comparison with the accuracies in the top row (e.g. for some rules, the zero \Delta accuracy for FLAN-T5 is due to zero absolute accuracy).”
>
>   Additionally, we will provide OOD charts with absolute proof accuracies in the Appendix.
>
>
> 4. “Different from previous works, this paper claims that LLMs can generalize compositionally. Can you further compare your work with them, and analyze why different conclusions are made?”
>
> We will add the following sentence to Section 4.2.2 on line 186: “But there is prior work showing that models can generalize compositionally in some settings, such as in [1] (see Figure 4) and in [2] (see Figure 6).”
>
>   We emphasize that much of the prior work showing models are unable to generalize compositionally was done on semantic parsing and not on reasoning, and so our findings do not directly contradict with that body of work. We are also careful to disentangle composition with respect to rules vs proof depth, which are often conflated. We find in new experiments that LMs are not able to find proofs with very large depth. The experiments are described in greater detail in (1) in the response to Reviewer w9uz, with results shown in the new figures in the 1-page supplementary PDF.
>
> [1] Arian Hosseini, Ankit Vani, Dzmitry Bahdanau, Alessandro Sordoni, and Aaron C. Courville. On the compositional generalization gap of in-context learning. BlackboxNLP@EMNLP 2022. URL https://aclanthology.org/2022.blackboxnlp-1.22.
>
> [2] Ofir Press, Muru Zhang, Sewon Min, Ludwig Schmidt, Noah A. Smith, & Mike Lewis (2022). Measuring and Narrowing the Compositionality Gap in Language Models. CoRR, abs/2210.03350. URL https://arxiv.org/abs/2210.03350.
>
> 5. “I cannot understand the subtitle of section 4.1. [Maybe] a more concise and academic title would be better.”
>
> We thank the reviewer for the suggestion. We will rename the section to be clearer.

---

> > ### Comment · Reviewer_ReKG · 2023-08-16
> >
> > Thanks for the response. Most of my major concerns are well addressed. I will raise my score to 7.
> >
> > - “Our findings suggest that in-context learning is best applied to ... to overfitting.”
> > I am curious whether this is true in practice. Some experiments will significantly improve the value of the claim as well as the paper.

---

### Official Review · Reviewer_w9uz · 2023-07-07

**Soundness:** 3 good
**Presentation:** 3 good
**Contribution:** 3 good
**Rating:** 6
**Confidence:** 4

**Summary:**

The authors propose a new dataset ProntoQA-OOD, to evaluate LLMs' reasoning abilities across more complex proofs across multiple aspects such as depth, compositionality, etc. The authors find that LLMs are able to generalize to longer and compositional proofs but usually require demonstrations to get these good accuracies.

**Strengths:**

* The analysis done in the paper is pretty comprehensive and contains some interesting findings that can help future LLM research.
* The dataset is an interesting resource for future reasoning-based evaluation of LLMs

**Weaknesses:**

* The main weakness is the lack of meaningful insights from the results, which are mostly contradictory to previous findings. Since the majority of the observations contradict previous works (section 4.2.1, 4.2.2, 4.3), its not sufficient to just state the observation and note its in stark contrast to previous findings. It'd help to take few of these scenarios and deep dive on why the observations are so different. Likely reasons could be the kind of datasets, number of ICL examples, etc. Please add a case study comparing these models across previous datasets and this dataset to make the analysis more interesting.
* Another issue with so many surprising results where the authors claim that LLMs are good at a task (which previous papers found to be hard to solve), leads to the question about the specific data distribution being used to test these models. Maybe the data is very basic and in-distribution (i.e., the models already have strong knowledge about these instances). Comparisons with a synthetic dataset such as ProofWriter can help alleviate this issue.
* Observations made by the authors in Section 4.2.2 (lines 192-193) that its not always best to provide demonstrations from the same distribution is in contrast to the general claim that LLMs do require in-demonstration examples (contribution 3). More discussion on this point needs to be done to explain when some examples are helpful and when they are harmful.
* The work on expanding compostionality is very similar to a prior work [1] that explores OOD compositonality. Consider discussing that in the related works.

[1] RobustLR: A Diagnostic Benchmark for Evaluating Logical Robustness of Deductive Reasoners, EMNLP 2022

**Questions:**

See weakness section for suggestions.

---

> ### Author Rebuttal · Authors · 2023-08-10
>
> Thank you for the feedback and helpful comments! We would greatly appreciate any additional comments.
>
> 1. “The main weakness is the lack of meaningful insights from the results, which are mostly contradictory to previous findings…”
>
> In addition to the summary of contributions in the common rebuttal, we ran a handful of new experiments to perform a more in-depth investigation into some of the observations.
>
>   For the experiments probing the generalization capacity of LMs on proofs with greater depth and width, we were previously limited by the context limit of the LMs and were unable to test for higher proof depth and width values. Thus, we performed new experiments where we reduced the number of in-context (IC) examples from 8 to 4 and tested for greater proof depths and widths. We replace Fig 7 and 8 with Fig 12 in the supplementary PDF and rephrase Sec 4.2.3:
>
>   “As is evident from [New Fig], when shown demonstrations of proofs of depth 2, the models’ performance decreases with increasing depth. But this is due to the increase in the inherent difficulty of the task, as both ID and OOD accuracies decrease with increasing depth. Though the notable exception is GPT-3.5 on conjunction elimination, where ID accuracy remains high as OOD accuracy decreases. Models are able to generalize better with increasing proof width on conjunction elimination, but only GPT-3.5 is able to generalize to greater proof widths on conjunction introduction.”
>
>   Having probed higher depths, these results are more consistent with previous findings, and helps to disentangle the effects of proof width, depth, and rule composition.
>
> Taking a closer look at Fig 9, we observe for some deduction rules, removing distractors from the IC examples improved performance on test examples with distractors. We ran a new experiment, shown in Fig 13 of the supplementary PDF, where we replaced each distractor in the IC examples with an irrelevant sentence, unrelated to the ground truth proof. This tests whether the increased length of each IC example could cause poorer ID performance than OOD. We replace Fig 9 with 13 and add the following to Sec 4.3:
>
>   “The bottom row of [New Fig] shows the result where each distractor sentence in the IC examples is replaced with an irrelevant sentence. Irrelevant sentences are unrelated to the ground truth proof, containing entirely distinct entities and concepts. We observe that replacing the distractors with irrelevant sentences either does not change or decreases the accuracy, suggesting that, for conjunction introduction, the improved accuracy in the OOD setting is likely due to the fact that each IC example is not as long as in the ID setting. Thus, at least for some deduction rules, the model may be better primed for the task by simpler demonstrations than by those that are ID.”
>
>   To address the concern with Sec 4.2.1, we replace the sentence on line 172: “This is in contrast with McKenzie et al. [2023] which showed that reasoning with modus tollens exhibited inverse scaling behavior... More recently, [1] has shown that when trained with additional compute, models are able to perform modus tollens.”
>
>   We also note that the modus tollens task in the Inverse Scaling Prize is distinct from our task, and the results are not directly comparable.
>
> We emphasize that much of the prior work showing models are unable to generalize compositionally was done on semantic parsing and not on reasoning, and so our findings complement but do not directly contradict with that body of work.
>
>   We add the following to Sec 4.2.2 on line 186: “But there is prior work showing that models can generalize compositionally in some settings, such as in [2] (see Fig 4) and in [3] (see Fig 6).”
>
> [1] Wei et al. 2022. Inverse scaling can become U-shaped.
>
> [2] Hosseini et al. 2022. On the compositional generalization gap of in-context learning.
>
> [3] Press et al. 2022. Measuring and narrowing the compositionality gap in language models.
>
>
> 2. The data distribution used to test the models may be too simple.
>
> Our dataset is synthetic and specifically uses fictional names for concepts to avoid any confounding effects from the model’s knowledge acquired during pretraining. Compared with ProofWriter, our dataset tests for many more rules of deduction (covering all logical connectives in propositional logic), whereas ProofWriter tests only for modus ponens and conjunction rules. [1] demonstrate that the previous version of PrOntoQA is comparable with ProofWriter in terms of difficulty (compare Fig 2b and c in [1]), and the set of generated questions of PrOntoQA-OOD is a strict superset of that of PrOntoQA.
>
> Fig 12 of the supplementary PDF shows a new experiment that probes the model’s ability to find proofs of depth much greater than that shown in Fig 7. The models do indeed struggle with proofs of such depth, indicating that the data distribution is likely not in the training distribution of the models.
>
> [1] Kazemi et al. 2023. LAMBADA: Backward chaining for automated reasoning in natural language.
>
>
> 3. “More discussion on this point needs to be done to explain when some [IC] examples are helpful and when they are harmful.”
>
> We agree and have added discussion as described in the common rebuttal. Specifically, we make a distinction between deduction rules with which the model is more vs less familiar. Our results show that in-demonstration examples are required for proof by cases and proof by contradiction, but not for other deduction rules.
>
>
> 4. “The work on expanding compositionality is very similar to a prior work [1] that explores OOD compositionality. Consider discussing in related work”
>
> We thank the reviewer for bringing this work to our attention. They similarly use synthetic examples to test the OOD reasoning capacity of models after fine-tuning. We will add it to the paragraph beginning on line 65.
>
> [1] Sanyal et al. 2022. RobustLR: A diagnostic benchmark for evaluating logical robustness of deductive reasoners.

---

> > ### Comment · Reviewer_w9uz · 2023-08-13
> > **Response to authors**
> >
> > Thanks for conducting these experiments quickly and adding more insights. Both the findings about proof depth vs width and addition of redundant distractors are interesting. These points should be highlighted not just in the relevant sections, but also in introduction, and possibly conclusions. I'm happy to increase my scores to reflect my new position about the paper.

---

### Author Rebuttal · Authors · 2023-08-10

Thank you to all reviewers for the very helpful feedback and comments! We will address common concerns below and also respond to each reviewer individually.

1. Some reviewers have raised the concern that there is a lack of meaningful insights from the results, especially those that are contradictory with previous findings.

The following is brief summary of the main contributions and insights in our paper:
 - We believe ours is the first comprehensive exploration of LMs’ reasoning ability on a complete set of deduction rules (covering all logical connectives of propositional logic). Our proposed testbed provides a way to evaluate reasoning ability on deduction rules that are not covered by existing natural language benchmarks, and we believe this is valuable to the field.
 - We disentangle “composition” into three types–rule composition, depth, and width–and test LMs’ generalization ability with respect to each type. Previous work often conflated depth generalization with generalization via rule composition [1,2], and we find that LMs are able to generalize compositionally in reasoning in terms of rule composition, while proofs of larger depth are still challenging.
 - LMs are less familiar with proof by cases and proof by contradiction than with other deduction rules. Thus any task that requires reasoning over disjunction and negation would benefit from explicit demonstrations of those deduction rules, as in our experiments. In addition, it provides an explanation for why LMs struggle on tasks involving negation [3,4]. This finding can inform future pretraining or finetuning of models to improve their knowledge of these rules.
 - In some settings, we find LMs perform better on tasks when the in-context examples are not from the same distribution as the test example. We observe via new experiments that longer demonstrations can be harmful. This raises new questions to pursue, to further characterize better distributions for in-context demonstrations and to understand the mechanism of learning from in-context examples.

Some of these points are supported by new experiments which we describe in detail in (1) in the response to Reviewer w9uz. The results of these experiments are shown in new figures in the 1-page supplementary PDF.

[1] Hanlin Zhang, Yi-Fan Zhang, Li Erran Li, & Eric P. Xing (2022). The Impact of Symbolic Representations on In-context Learning for Few-shot Reasoning. CoRR, abs/2212.08686. URL https://arxiv.org/pdf/2212.08686.pdf.

[2] Nicolas Gontier, Siva Reddy, & Christopher Pal (2022). Does Entity Abstraction Help Generative Transformers Reason?. Trans. Mach. Learn. Res., 2022. URL https://arxiv.org/pdf/2201.01787.pdf.

[3] RobustLR: A Diagnostic Benchmark for Evaluating Logical Robustness of Deductive Reasoners, EMNLP 2022. URL https://aclanthology.org/2022.emnlp-main.653.pdf.

[4] Thinh Hung Truong, Timothy Baldwin, Karin Verspoor, & Trevor Cohn (2023). Language Models are not Naysayers: An Analysis of Language Models on Negation Benchmarks. *SEM@ACL 2023. URL https://arxiv.org/abs/2306.08189.


2. Lack of insights/recommendations on how to best use in-context learning for reasoning, given our findings.

We address this concern by adding the following sentences to the Introduction (and the Conclusion, space permitting):

  “Our findings suggest that in-context learning is best applied to reasoning tasks by including examples that cover a diverse set of deduction rules, and keeping the examples simple. The in-context examples should especially contain examples of deduction rules that are less familiar to the model (i.e. proof by cases and proof by contradiction), and distractors should be provided for such examples as the model is more prone to overfitting.”

---

### Decision · Program_Chairs · 2023-09-21

**Decision:**

Accept (poster)

**Comment:**

This paper studies the deductive inference capabilities of LLMs in a way that is more thorough than prior work. Many interesting insights are extracted from the experiments, around compositionality vs. reasoning depth, etc.